# The Use of Compost Tea in a Containerized Urban Tree Nursery Shows No Evident Benefits to Tree Growth or Mycorrhizal Colonization

Dan Du [1],*, Stephen J. Livesley [1] , Stefan K. Arndt [1] , Camille Truong [2] and Rebecca E. Miller [1,2]

1    School of Agriculture, Food and Ecosystem Sciences, The University of Melbourne, Richmond, VIC 3121, Australia; sjlive@unimelb.edu.au (S.J.L.)
2    Royal Botanic Gardens Victoria, Melbourne, VIC 3004, Australia
*    Correspondence: rebeccadd@gmail.com

**Abstract:** Compost tea is a liquid organic amendment that has been reported to benefit plant growth and performance through positive effects on microbial communities and plant nutrition. However, few studies have demonstrated this for containerized plants produced in tree nurseries. Five common urban tree species (*Acer negundo*, *Corymbia maculata*, *Ficus platypoda*, *Hymenosporum flavum*, *Jacaranda mimosifolia*) were grown in a containerized experiment to investigate the effects of compost tea application on tree growth and root mycorrhizal colonization over six months. The microbial composition of compost tea was also determined with 16S (bacteria) and ITS1 (fungi) metabarcoding. No significant positive effect of compost tea on plant growth or root mycorrhizal colonization was observed. Roots of all tree species were colonized by one type of mycorrhizal fungi, ectomycorrhizae (ECM), or vesicular–arbuscular mycorrhizae (VAM). However, no relationship between the mycorrhizal colonization percentage and plant growth was detected. Thus, there was no evidence that a once-off application of compost tea had benefits for mycorrhizal colonization and growth of containerized trees in a nursery setting. Further research is needed to investigate whether any benefit from compost tea is evident once containerized trees are planted into urban landscapes where growth conditions may be more challenging.

**Keywords:** nutrient; fertilizer; mycorrhizae; ectomycorrhizae; endomycorrhizae; vesicular; arbuscular

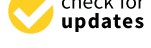



## 1. Introduction

Compost tea is an organic soil amendment that is gathering attention due to its reported benefits and positive effects on plants due to its direct nutrient value [1–4] and the promotion of microbial communities in soil or artificial substrates [2,3,5–9]. These improved microbial communities can further promote plant performance by improving soil properties [10], recycling nutrients [11,12], defending against pests and pathogens [1,8,13], and promoting mutualistic associations with plants [2].

Root symbiosis with mycorrhizal fungi is one type of microbial association that has been observed in 75%–90% of terrestrial plants studied so far [14]. Mycorrhizae can be categorized into four types based on their morphology and the fungal and plant groups involved: ectomycorrhizae (ECM) and vesicular–arbuscular mycorrhizae (VAM), which are found in a wide diversity of plant taxa, and ericoid and orchid mycorrhizae, which are limited to members of Ericaceae and Orchidaceae, respectively [14,15]. Mycorrhizal associations provide multiple benefits to plants, including enhanced uptake of water and nutrients [10,16], and protection against both abiotic and biotic stresses, such as heavy metals [17–19], salinity [20,21], and soil pathogens [22,23]. They can also improve soil structure by contributing to the formation of long-lasting soil aggregates [10], and even increase plant root longevity [22].

All these benefits from mycorrhizal fungi are of particular importance to trees that are planted and grown in potentially harsh urban soil conditions, which may include water stress, nutrient deficiency, and compaction [24]. Furthermore, trees grown in urban environments often have lower levels of mycorrhizal colonization and are associated with fewer fungal species compared to trees grown in natural forest landscapes [25–27]. It is unknown if compost tea application during the standard tree nursery production process promotes both tree growth rates and root mycorrhizal associations. If compost tea is facilitating mycorrhizal associations, then those trees could be better equipped when they are planted into harsh urban soil environments.

Studies investigating effects of compost tea on plants adopt a single application [28–30] or multiple regular applications [2,4,9,31], in experiments conducted over weeks to months. To date, most studies working with compost tea and/or mycorrhizal fungi have focused on agricultural crops [2,4,8,28–30,32] and very few have investigated similar applications in the horticultural industry [13], including for advanced tree production in a nursery setting [6,33]. Further, few studies have analysed the microbial composition of compost tea [2,6] and whether it contains microorganisms to facilitate potential positive (e.g., mycorrhizae) or negative (e.g., pathogens) effects on tree growth both in the short term (during nursery production) and long term (once planted in the landscape). Working in collaboration with an advanced tree nursery located near Melbourne (VIC, Australia), this study aimed to investigate the microbial composition of their compost tea and the effects of a once-off application of such compost tea to promote mycorrhizal association with tree roots and tree growth, in a standard containerized nursery production system following standard nursery operation practice. Specifically, we aimed to test the following hypotheses:

- Compost tea application will have a positive effect on containerized tree growth rates;
- Compost tea application will have a positive effect on root mycorrhizal colonization percentages;
- Percentages of root mycorrhizal colonization will positively correlate with rates of tree growth.

## 2. Materials and Methods

### 2.1. Compost Tea Formula

The compost was prepared according to the nursery's standard practice with ingredients from both green wastes produced within the nursery premises and commercially purchased products (Table 1). Root ball material was added specifically as a potential source of mycorrhizal inoculum. A large (5 kg) compost tea bag was made from a grain bag used in the beer brewing industry. The compost tea bag was placed into a 200 L barrel containing 175 L of water, 1.5 L of fish hydrolysate, 1.5 L of cold-pressed kelp, and 1.5 L of molasses. A water pump was used to create a vortex and to provide aeration at the same time, for approximately 24 h. The compost tea was then added to containerized trees on the same day.

### 2.2. Focal Species

We selected five tree species belonging to five different plant families, including native Australian and exotic species [34], as well as deciduous and evergreen species (Table 2). The type of their mycorrhizal associations (VAM and/or ECM) was defined according to Soudzilovskaia et al. [35]. The functioning of mycorrhizae in these five species and the main constraints in artificial substrates and urban soils are currently unclear and rarely reported.

### 2.3. Experimental Design and Maintenance

For each tree species, 10 trees of similar size were selected for the experiment ($n = 50$). All the trees used in this study were between 1 and 3 years old when the experiment started. They were randomly assigned into either the control ($n = 5$) or compost tea treatment ($n = 5$) groups. The substrate used for this experiment was made by the nursery with

composted pine bark, wood fibre, humate, and sand (Table S1) with air-filled porosity between 18%–21%. The compost tea was added to the treatment group in early spring on 8 September 2020. Based on the plant size, *Acer negundo* was kept in 27 L pots, and the other four species were kept in 14 L pots. The compost tea was applied to the treatment group based on growing substrate volume in each pot, approximately 20 mL per L of growing substrate. The growth of all 50 trees, treated and control, was closely monitored and the trees were maintained according to nursery standard practices for six months. All 50 containerized trees were fertilized every three months with 2 g of inorganic fertilizer (N:P:K = 9:4:12) per L of substrate applied as top dressing to each pot. Manual weeding was conducted during the experiment when needed. Structural pruning was used to maintain a dominant leader. Daily irrigation was provided to each container via a drip line.

**Table 1.** Ingredients for compost used for this study. The core temperature of the compost pile was closely monitored to prevent overheating. Once the temperature reached 60–75 °C, the pile was manually turned. After it was turned four times, the pile was left to mature for two months. The compost was used to make compost tea after maturation.

| Ingredient | Amount (by Volume) | Additional Information |
| --- | --- | --- |
| Leaf trimmings | 10%–15% | Green waste on site |
| Weeds | 20%–25% | Green waste on site |
| Grass clippings | 5%–10% | Green waste on site |
| Dynamic lifter | 5% | Commercial product |
| Horse manure | 5% | - |
| Seamungus | 5% | Commercial product |
| Rock dust | 2% | - |
| Wood chips | 25%–30% | - |
| Root balls | 10%–15% | From discarded mixed tree species on site |
| Mature compost | 5% | From an old compost pile made previously in the same way |

**Table 2.** Tree species included in the experiment, including information on their mycorrhizal types, i.e., vesicular–arbuscular mycorrhizae (VAM) and/or ectomycorrhizae (ECM).

| Family | Species | Type | Origin | Mycorrhizal Type |
| --- | --- | --- | --- | --- |
| Sapindaceae | *Acer negundo* L. | Deciduous | Exotic | VAM |
| Myrtaceae | *Corymbia maculata* (Hook.) K.D. Hill & L.A.S. Johnson | Evergreen | Native | Dual (VAM/ECM) |
| Moraceae | *Ficus platypoda* (Miq.) A. Cunn. ex Miq. | Evergreen | Native | VAM |
| Pittosporaceae | *Hymenosporum flavum* F. Muell. | Evergreen | Native | VAM |
| Bignoniaceae | *Jacaranda mimosifolia* D. Don | Deciduous | Exotic | VAM |

### 2.4. Plant Growth Measurements

To investigate any effect of compost tea treatment on plant growth, tree growth was measured as stem height and stem diameter for six months after the compost tea application (from 8 September 2020 in spring to 19 March 2021 in autumn). Studies working with compost tea and/or mycorrhizal associations vary greatly in experiment duration, ranging from weeks to months [2,6,21]. Furthermore, studies on mycorrhizae indicate that colonization after exposure to inoculum occurs within 2–6 weeks [36]. Six months would therefore be long enough for mycorrhizal associations to have developed and plant growth response to be evident. Stem height above substrate surface in the container was measured using a 5 m measuring tape, accurate to 1 cm. The maximum stem diameter at 30 cm above the substrate surface in the container was measured using a digital Vernier calliper, accurate to 0.01 mm. Initial measurements of tree size (Table S2) were taken on 14 September 2020, six days after compost tea treatment. Measurements were made on eight occasions during the six-month experiment, with final measurements taken on 19 March 2021. The relative growth rates (RGRs) (in mm mm$^{-1}$ day$^{-1}$) were calculated according to Equations (1) and (2). All measurements were transformed into mm, and

the time period (T) was in days; $H_0$ and $D_0$ indicate the initial stem height and diameter, respectively, and $H_t$ and $D_t$ are the final stem height and diameter.

$$\text{RGR of Height} = \frac{\log(H_t) - \log(H_0)}{T} \tag{1}$$

$$\text{RGR of Diameter} = \frac{\log(D_t) - \log(D_0)}{T} \tag{2}$$

*2.5. Root Sample Collection and Processing*

Root samples were collected and processed four months after compost tea application, from 18–28 January 2021. A steel corer with inner diameter of 4 cm was used to collect one core containing substrate and fine roots from each container, except for *C. maculata*, which required two cores per container. The corer was cleaned with water between sampling of each container. The cores were placed into Ziplock bags and stored in a chilled cool box for transport to the laboratory. In the laboratory, core samples were soaked in water for approximately 30 min and fine roots were separated from surrounding substrate particles. Each core sample was sufficient to obtain >2 g fresh weight of fine roots. The cleaned root samples were preserved in 70% ethanol solution in water (*v/v*) within 24 h of collection, prior to mycorrhizal colonization assessment.

*2.6. Assessment of Mycorrhizal Colonization*

To quantify root mycorrhizal colonization, the clean root samples were processed using a modified ink and vinegar method developed by Vierheilig et al. [37]. The roots kept in 70% ethanol water solution (*v/v*) were rinsed with distilled water, then cleared by soaking in a 10% KOH solution at room temperature. The clearing times differed among species, from 4 days to 3 weeks. Cleared roots were rinsed in acidified distilled water (containing a few drops of white vinegar) and dyed with a 5% ink–vinegar solution overnight at room temperature. Excess stain was removed by washing roots in diluted white vinegar (5%). Roots were stored in 50% glycerol solution in water (*v/v*) for microscopic observation. Mycorrhizal colonization was assessed following the gridline intersection method [15,38] (Figure S1). VAM and ECM fungal colonization percentages were assessed separately on a random subsample of stained roots, based on key features, including vesicles, arbuscules, and nonseptate hyphae as diagnostic structures for VAM, and fungal mantle and Hartig net for ECM [15]. When a piece of root crossed a gridline, the presence or absence of mycorrhizae at that intersection was noted based on the identifying features observed. At least 100 intersections per subsample were counted to calculate the percentage of mycorrhizal colonization (Equation (3)).

$$\text{Mycorrhizal colonization percentage} = \frac{\text{Intersection count with mycorrhizae presence}}{\text{Total intersection count}} \times 100\% \tag{3}$$

*2.7. Microbial Community Profiling of Compost Tea*

To verify the presence of mycorrhizal fungi in the compost tea, and potential for pathogen introduction, we characterized the bacterial and fungal communities present in the compost tea using DNA meta-barcoding. A sample was frozen at −80 °C and sent to the Australian Genome Research Facility (AGRF) for DNA extraction and amplicon sequencing using Illumina MiSeq 2 × 300 bp. Briefly, DNA was extracted using the DNeasy PowerLyzer Power Soil Kit (Qiagen), and the V3–V4 region of bacterial 16S rDNA was amplified by PCR with primers 341F/806R; for fungi, the ITS1 region of nuclear rDNA was amplified with primers ITS1F/ITS2 [39]. Amplicons were generated using the Platinum SuperFi II mastermix (Life Technologies, Australia) with the following conditions: Initial denaturation at 98 °C for 30 s, followed by 30 cycles of 98 °C for 30 s, 60 °C for 10 s, and 72 °C for 30 s, with a final elongation step at 72 °C for 5 min. Because our data did not include control samples to account for potential cross-contamination during library

preparation and multiplexing, we used stringent quality filters and only considered taxa with relative abundance of >0.5% of the total read count in the sample. Raw reads were quality-filtered in Trimmomatic [40] (SLIDINGWINDOW:4:30, MINLEN:100), followed by primers/adaptors trimming, denoising, and OTU clustering at 97% similarity using AMPtk [41]. Taxonomy assignment of OTUs was performed using a "hybrid" approach in AMPtk against the SILVA 138.1 (16S) and UNITE v.9.0 (ITS) reference databases, and fungal functional guilds were assigned according to FungalTraits [42].

### 2.8. Data Analysis

Data analyses were performed using Rstudio (R version 4.2.1 and packages including tidyr, dplyr, ggpubr, and performance) [43]. The main three numeric variables, RGRs of stem height and diameter, and percentage of mycorrhizal colonization, were checked for normal distribution using the Shapiro–Wilk test, and homogeneity of variance was also checked using the performance package of R. In order to satisfy the general linear model (GLM) assumptions, the mycorrhizal colonization percentage was logistic-transformed using the formula $\log(X/(1-X))$. Multifactor GLMs were then used to investigate (1) the effects of treatment and species on RGRs of stem height and diameter and (2) the effects of treatment and species on mycorrhizal colonization percentage. Linear regression models were used to investigate the relationships between mycorrhizal colonization percentage and plant growth rates.

## 3. Results

### 3.1. Plant Growth Response to Compost Tea Treatment

There was no significant positive effect of compost tea treatment on either the RGR of stem height or stem diameter (Figure 1). The exception was *A. negundo*, where compost tea treatment had a significant ($p = 0.03$) negative effect on the RGR of stem diameter (Figure 1B). There were significant interspecific differences in RGR of stem height ($p < 0.001$) but not in RGR of stem diameter ($p = 0.676$).

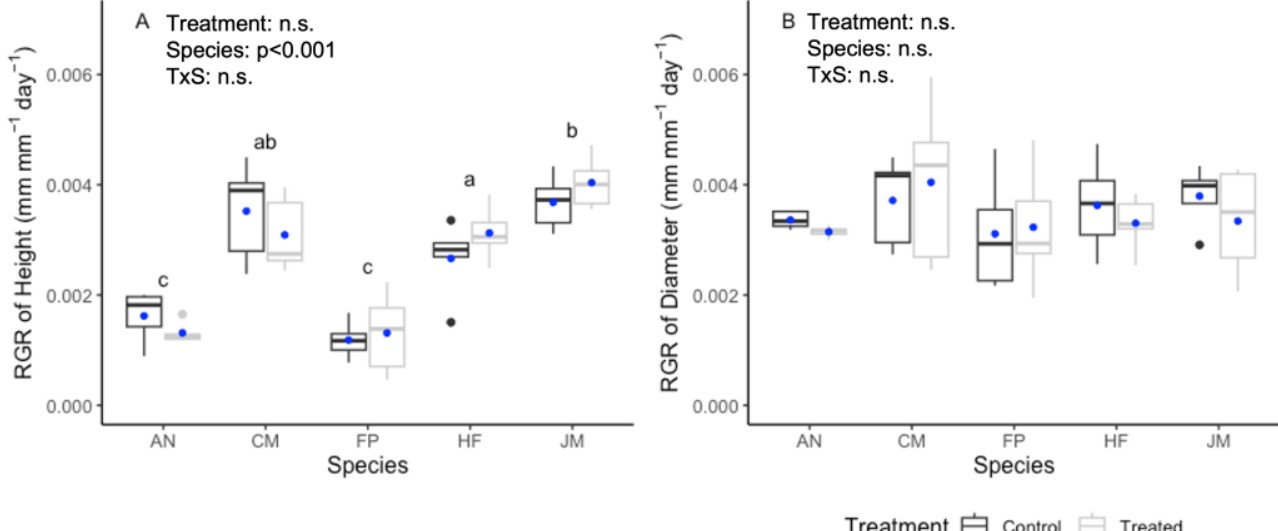

**Figure 1.** Relative growth rates (RGRs) of (**A**) stem height and (**B**) stem diameter at 30 cm above the substrate level for five tree species for six months after compost tea treatment (treated) or no treatment (control). Tree species are *Acer negundo* (AN), *Corymbia maculata* (CM), *Ficus platypoda* (FP), *Hymenosporum flavum* (HF), and *Jacaranda mimosifolia* (JM). For each box plot (*n* = 5 trees), the top and bottom boundaries are the third and first quartiles, the middle line is the median, and the blue dot is the mean. The light and dark grey dots are outliers. Results of 2-factor GLMs are shown; different letters indicate significant differences at *p* < 0.05 between species determined using Tukey's HSD post hoc test.

### 3.2. Root Mycorrhizal Colonization Response to Compost Tea Treatment

The roots of all 50 trees were colonized with mycorrhizal fungi regardless of treatment. The compost tea treatment had no overall significant positive effect on mycorrhizal colonization percentage (Figure 2). The exception was *C. maculata*, where the compost tea treatment had a significant negative effect upon the mycorrhizal colonization percentage ($p = 0.022$). The extent of colonization ranged from 2% to 78% across all individual trees and varied between pots of the same species. Mycorrhizal colonization percentage differed significantly among species ($p < 0.01$), but there was no significant interaction between treatment and species. Additionally, no significant relationship was detected between mycorrhizal colonization percentage and plant growth for any species (Figure S2), with RGR of stem height ($p = 0.213$) or stem diameter ($p = 0.399$).

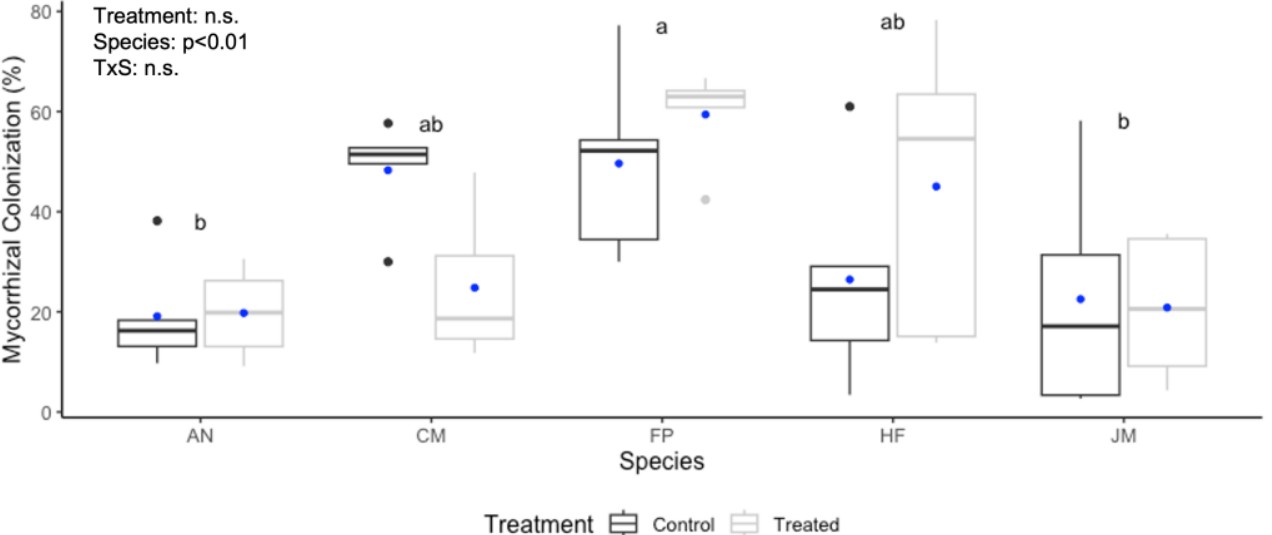

**Figure 2.** Mycorrhizal colonization percentage (%) of five tree species four months after compost tea treatment (treated) or no treatment (control). Tree species are *Acer negundo* (AN), *Corymbia maculata* (CM), *Ficus platypoda* (FP), *Hymenosporum flavum* (HF), and *Jacaranda mimosifolia* (JM). For each box plot ($n = 5$ trees), the top and bottom boundaries are the third and first quartiles, the middle line is the median, and the blue dot is the mean. The light and dark grey dots are outliers. Results of a 2-factor GLM are shown; different letters indicate significant differences at $p < 0.05$ between species determined using Tukey's HSD post hoc test.

### 3.3. Mycorrhizal Colonization Types

*Acer negundo*, *F. platypoda*, *H. flavum*, and *J. mimosifolia* formed VAM associations, with visible arbuscules, vesicles, and associated internal and external nonseptate hyphae (Figure S3). *Corymbia maculata* formed ECM associations, with visible mantles and dense hyphae, both internal and external (Figure S3); no evidence of dual VAM/ECM associations was observed for *C. maculata* roots.

### 3.4. The Bacterial and Fungal Composition of the Compost Tea

A total of 20,284 fungal reads (ITS1) passed quality filtering and clustered into 19 taxa (Table 3). Two taxa could only be identified at family level, in addition to one unknown fungus. The compost tea sample was mostly composed of saprotroph species (yeast, mould, soft rot, and filamentous fungi commonly found in soils), with some endophytic species and putative plant and animal pathogens. No mycorrhizal VAM or ECM fungal taxa were detected in the compost tea. The number of bacterial reads (16S) that passed quality filtering was 35,273. They clustered into 17 species that were mostly decomposers and potential pathogens (Table 4).

**Table 3.** Fungal taxa (read relative abundance > 0.5%) detected in the compost tea sample using ITS1 meta-barcoding, with their primary/secondary functions and lifeforms according to FungalTraits [42].

| ID | Primary Lifestyle | Secondary Lifestyle | Relative Abund. (%) | Additional Information |
|---|---|---|---|---|
| *Barnettozyma californica* | Saprotroph (various substrates) | | 25.07 | Yeast, sugar-rich substrates |
| *Mortierella indohii* | Soil saprotroph | Root-associated | 14.53 | Filamentous mycelium |
| *Penicillium* sp. | Saprotroph (various substrates) | Foliar endophyte | 11.82 | Mould, some species are toxin-producing, animal parasites or mycoparasites |
| *Chaetomium piluliferum* | Litter saprotroph | Foliar endophyte | 9.44 | Soft rot |
| *Trichothecium roseum* | Plant pathogen | Litter saprotroph | 8.60 | Filamentous mycelium |
| *Thermomyces* sp. | Soil saprotroph | Litter saprotroph | 5.64 | Mould, potential plant pathogenicity |
| Unknown Trichosporonaceae | – | – | 4.16 | – |
| *Rhodotorula* sp. | Saprotroph (various substrates) | Foliar endophyte | 3.89 | Yeast |
| *Candida* sp. | Saprotroph (various substrates) | | 3.62 | Yeast, sugar-rich substrates |
| *Byssochlamys* sp. | Saprotroph (various substrates) | | 3.13 | Mould, some species are food spoilage agents |
| *Mortierella reticulata* | Soil saprotroph | Root-associated | 2.52 | Filamentous mycelium |
| Unknown Fungi | – | – | 2.22 | – |
| *Fusarium oxysporum* | Plant pathogen | Litter saprotroph | 1.24 | Soft rot |
| *Mortierella* sp. | Soil saprotroph | Root-associated | 1.00 | Filamentous mycelium |
| Unknown Nectriaceae | – | – | 0.76 | – |
| *Aspergillus fumigatus* | Saprotroph (various substrates) | Foliar endophyte | 0.67 | Mould |
| *Cutaneotrichosporon* sp. | Animal parasite | Animal decomposer | 0.58 | Yeast |
| *Wickerhamomyces anomalus* | Litter saprotroph | Saprotroph | 0.57 | Yeast, sugar-rich substrates |
| *Acremonium* sp. | Saprotroph (various substrates) | Foliar endophyte | 0.56 | Soft rot, potential plant pathogenicity |

**Table 4.** Bacterial taxa (read relative abundance > 0.5%) detected in the compost tea sample using 16S meta-barcoding, with their function, relative abundance (%), and additional information for each species listed based on cited reports from the literature.

| ID | Function | Relative Abund. (%) | Additional Information |
|---|---|---|---|
| *Weissella* sp. | Probiotic/pathogen | 25.60 | Prolific in environment, probiotic or pathogenic [44] |
| *Acinetobacter* sp. | Pathogen/degrader | 22.27 | Can be pathogenic to humans [45] |
| *Acinetobacter guillouiae* | Unknown | 11.05 | An environmental species [46] |
| *Pseudomonas* sp. | Pathogen/degrader | 10.30 | Commonly exist in soil and can be plant pathogens [11] |
| Unknown Enterobacteriaceae | Unknown | 4.15 | |
| *Pseudomonas veronii* | Degrader | 3.52 | A bioremediation of contaminated soils [47] |
| *Acinetobacter* sp. | Pathogen/degrader | 3.50 | Can be pathogenic to human [45] |
| *Enterobacter* sp. | Probiotic | 3.36 | Might be nitrogen fixing bacteria [48] |
| *Acetinobacter* sp. | Pathogen/degrader | 3.32 | Can be pathogenic to human [45] |
| *Pseudomonas* sp. | Pathogen/degrader | 3.22 | Commonly exist in soil and can be plant pathogens [11] |
| *Hafnia* sp. | Pathogen | 2.14 | Opportunistic pathogen |
| *Rahnella aquatilis* | Pathogen | 1.67 | Opportunistic pathogen [49] |
| *Serratia* sp. | Pathogen | 1.67 | Opportunistic pathogen |
| *Bacillus* sp. | Unclear | 1.47 | Commonly exist in soil and might fix nitrogen [11] |
| *Lactococcus* sp. | Lactic acid bacteria | 1.04 | They are generally safe and produce lactic acid [50] |
| *Arthrobacter* sp. | Opportunistic pathogen | 0.87 | Commonly exists in soil [11] |
| *Acetinobacter* sp. | Pathogen/degrader | 0.86 | Can be pathogenic to human [45] |

## 4. Discussion

### 4.1. Plant Growth Response to Compost Tea

Our first hypothesis was that the compost tea would positively impact containerized plant growth. However, we did not measure a significant positive effect of compost tea treatment on growth in any of the five species (Figure 1). There are several possible explanations which relate to both the properties of the compost tea and the nursery production practices during this study. While compost tea can provide a direct source of nutrients to stimulate plant growth [1,4], it is unlikely that the single application of compost tea in this study would have stimulated growth due to any direct increase of nutrient supply. It has been shown that compost teas can contain low levels of macro- and micronutrients required for plant growth [1]. Consistent with that, Edenborn et al. [30] similarly reported no effect of a single compost tea application on eggplant growth, whereas a high frequency of application (weekly) of compost tea was able to promote pak choi plant growth in a containerized experiment [4]. In this study, we assessed how standard nursery practices affected plant growth which included a single application of compost tea as well as the application of inorganic fertilizer every three months. As such, it is unlikely that any small macro- or micronutritional benefit from the compost tea application would have been detected beyond the larger inorganic fertilizer benefit provided to both control and treated groups. Similarly, it is likely that any positive nutrient contribution from any enhanced microbial mineralization processes would have also been masked by the larger inorganic fertilizer benefits. In fact, the addition of inorganic nutrients, such as nitrogen, can impede soil microbial decomposition processes [51]. Studies have reported positive [2,4,6,8], negative [31,52], and no effects [30,33] from compost tea application upon plant growth of different plant species grown in various media. The relative growth rate of height varied greatly across the five studied tree species, and future studies should consider higher replication as this could test the significance of any effect of compost tea on tree stem height growth in a variety of species.

### 4.2. Mycorrhizal Colonization Response to Compost Tea Treatment

Our second hypothesis that compost tea would positively affect mycorrhizal colonization percentages was based on the observed positive effects of compost tea on soil microbiology [2,5,8]. This hypothesis assumes that, firstly, there may be some mycorrhizal fungi propagules (spores and/or hyphae) in the compost tea derived from composted root material and the environment, and, secondly, that other microbes, such as mycorrhizal "helper" bacteria, would support enhanced formation of mycorrhizal associations [53]. However, we did not observe a significant positive effect of compost tea treatment on mycorrhizal colonization percentage in any species, regardless of their type (VAM or ECM) of mycorrhizal association (Figure 2). Similar to our results, Ou-Zine et al. [32] reported no effect of compost tea on arbuscular mycorrhizal colonization in maize. In our study, trees from the control group showed similar and even higher percentage of mycorrhizal colonization compared with the treated group, which indicates that the standard nursery practices in place already allow for mycorrhizal colonization of these plants, even with the application of inorganic fertilizers, which can alter and inhibit mycorrhizal colonization [54,55]. Whether the significantly lower ECM colonization of *C. maculata* roots of plants treated with compost tea reflects the negative effect of compost tea on ectomycorrhizal colonization, through enhanced nutrient input, or antagonistic interactions among microbes [9] requires further investigation; however, highly variable mycorrhizal colonization of young eucalypt roots has been observed, even in field soils [56].

Importantly, we did not detect any mycorrhizal fungus in the compost tea (Table 3); therefore, it could not be a direct source of mycorrhizal fungal inoculum. It is to be noted that ITS1 and the primer set used may not be the most suitable to detect VAM that are usually studied using specific primers targeting the 18S or 28S regions, but ECM fungi are routinely detected using ITS1 [39]. Secondly, no known mycorrhizae helper bacteria were found in the compost tea. Additionally, it has to be recognized that DNA sequencing

identifies both living and dead genetic materials and therefore can indicate the presence but not necessarily functional activity of microbes in a sample.

Although studies have shown positive effects of compost teas in promoting microbial communities in various growing media [2,5,6,8,9], the outcomes of compost tea application can be very different, even contradictory, perhaps due to the lack of unified standards for compost tea [3]. Few of these studies have reported the specific fungal and/or microbial composition of the compost teas applied [2,6], and little is known about the interactions among microbes from compost tea and the ones existing in the growing media. Both positive synergistic [53] and antagonistic interactions [9] among the microbial communities have been observed. Although the meta-barcoding analyses revealed that the compost tea in this study contained beneficial bacteria and fungi that could be working as decomposers (Tables 3 and 4), it was not within the scope of this study to determine whether these probiotic microbes were established in the substrate after the one-time application. It may be that any nutritional and growth benefit from compost tea application is detectable once trees have been planted into the landscape, and nursery practices including fertilization cease.

### 4.3. Mycorrhizal Colonization and Plant Growth in a Container Production System

Our third hypothesis, that there would be a significant correlation between mycorrhizal colonization percentage and plant growth rate, was based on the well-known benefits of mycorrhizae for plant growth in both containerized and natural settings [10,16,25,57]. However, there was no evidence to support this hypothesis over the six months of this study. Perhaps most importantly, any plant growth response to mycorrhizal colonization might be impeded by existing nutrient availability in the growing media and by the inorganic fertilizer additions every three months. Nutrient levels, especially plant available phosphorus (P) concentrations in substrates, can significantly impact plant growth promotion by mycorrhizae [58–60]. Specifically, when the P level is sufficient, the contribution by mycorrhizal fungi gathering P for plants is insignificant, and growth responses to mycorrhizae are consequently more frequently reported under lower nutrients levels. For example, Wu and Xia [16] reported a significant positive effect of mycorrhizal colonization on plant growth and nutrient concentration in plant tissues in containerized citrus, with no fertilizer addition, while Sylvia et al. [61] reported no effect of mycorrhizal colonization on plant growth in 26 tree species when fertilizer was added, and Corkidi et al. [62] reported the same result in corn. Positive plant physiological and growth responses to mycorrhizal colonization are also more often reported under water deficit than well-watered conditions [63]. Given the regular fertilization and irrigation provided in this experiment, it is reasonable to conclude that beneficial mycorrhizal fungi activities were not realised, as no significant effect from mycorrhizal colonization on plant growth was identified.

In addition, the absence of any significant effect of mycorrhizal colonization percentage on plant growth rates may arise as different tree species can vary in their level of mycorrhizal dependency or responsiveness [64,65]. Little is known about mycorrhizal dependency and responsiveness of these and many other landscape tree species [61].

Even though our third hypothesis of a positive relationship between mycorrhizal colonization percentages and tree growth rates was rejected, there was also no evidence of a negative impact associated with mycorrhizal colonization. Unlike independent soil microbes, mycorrhizal fungi rely on plants to provide carbohydrates, but this usually comes at low cost to the plants that often release large amounts of excess carbon as exudates into the rhizosphere [66].

### 4.4. Other Effects of Compost Tea and Future Research

Despite the lack of a positive tree growth response or root mycorrhizal colonization percentage following compost tea application, there are several important findings for nursery container production systems. Firstly, all 50 trees in the experiment were colonized by mycorrhizal fungi, regardless of species, mycorrhizal types, or treatment. The good

levels of mycorrhizal colonization suggest that VAM and ECM inoculum is readily entering the nursery production system. According to the experiment setup, sources of mycorrhizal inoculum entering the tree pots include the substrate itself, the irrigation water, or simply airborne deposition from the surrounding vegetation and environment. Secondly, although trees were raised in an artificial medium in containers, they showed similar percentages of root mycorrhizal colonization to those reported for trees grown in natural soil [27,54].

While no benefits of compost tea application were detected in terms of plant growth, there may be other benefits for plants from root mycorrhizal colonization, such as defence against soil pathogens [22,23], but this was not part of our study. Studies on a range of agricultural species indicate that compost tea can assist with disease control [7,8,13] and pathogen suppression [5,29]. Although the composition of the compost tea used in this study included potential plant pathogens (Tables 3 and 4), we did not observe a negative effect of compost tea application on plant growth (Figure 1). The potential for improved disease resistance in tree production systems and further studies of the potential promotion of symbiotic associations [2] are worthwhile. Future research could include following the performance of trees from treated and nontreated groups that are planted in different urban landscapes (i.e., with degraded soil, lack of resources, and less diverse microbial communities) to investigate the impact of compost tea at the time of planting a containerized nursery tree into the harsh urban soil landscape. The nutritional needs of the trees under such conditions and the role of microbial communities, especially mycorrhizal fungi, in stress mitigation and resource acquisition could also provide valuable information for urban forest managers [67]. Future research could also investigate the interaction between mycorrhizal fungi naturally occurring in the soil with the microbial communities of compost tea, from the perspective of plant benefits, given that we detected several potential pathogens (Tables 3 and 4) and that antagonistic effects between the two have been determined in certain conditions [9]. The compost tea used for this study did not have any mycorrhizal fungal inoculum (Table 3), despite the addition of tree root material as one compost ingredient. Future research could explore other methods of effective mycorrhizal inoculation and their impact on tree nursery production and seedling performance [68,69]. Particular care should also be taken regarding introducing exotic microbial species into the environment when inoculating plants with artificial or food waste sources. Future research could also investigate the mycorrhizal dependency and responsiveness of tree species in urban environments. Given that trees grown in urban settings generally have lower mycorrhizal colonization percentages and fewer colonizing fungal species compared with their counterparts in natural forests [27], such knowledge would provide directions for advanced tree production and scientific guidance for improving the urban environment for tree establishment.

**Supplementary Materials:** The following supporting information can be downloaded at: https://www.mdpi.com/article/10.3390/f14061195/s1, Table S1: Ingredients for substrates; Table S2: Initial measurement of experiment trees; Figure S1: Method for mycorrhizal colonization percentage measurement; Figure S2: Scatterplots of mycorrhizal colonization percentage and plant growth; Figure S3: Images of roots colonized by mycorrhizal fungi in this study.

**Author Contributions:** Conceptualization, R.E.M. and D.D.; methodology, R.E.M. and D.D.; formal analysis, D.D. and C.T.; investigation, D.D.; resources, R.E.M. and S.J.L.; data curation, D.D.; writing—original draft preparation, D.D.; writing—review and editing, R.E.M., S.J.L., S.K.A., C.T. and D.D.; visualization, D.D.; supervision, R.E.M., S.J.L. and S.K.A.; funding acquisition, R.E.M. and D.D. All authors have read and agreed to the published version of the manuscript.

**Funding:** This research received no external funding.

**Data Availability Statement:** Not applicable.

**Acknowledgments:** The authors wish to thank Speciality Trees Nursery for the kind support, including all plants, media, the compost tea, and management of the experimental plants on their site. We also thank Hamish Mitchell, Ryan Bell, Sascha Andrusiak, Rowan Berry, Lisa Wittick, Vicky Waymouth, and Genevieve Hehir for their technical support. We thank Bihan Guo, Paul Cheung, Claire Kenefick, and Lavinia Chu for their volunteering help during the experiment. This research received support from the Madeleine Selwyn-Smith Memorial Award and Betty Elliott Horticulture Scholarship.

**Conflicts of Interest:** The authors declare no conflict of interest. The funders had no role in the design of the study; in the collection, analyses, or interpretation of data; in the writing of the manuscript; or in the decision to publish the results.

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
