# Peer review of "The Use of Compost Tea in a Containerized Urban Tree Nursery Shows No Evident Benefits to Tree Growth or Mycorrhizal Colonization"

_forests, doi:10.3390/f14061195_

Round 1
Reviewer 1 Report
« The use of compost tea in a containerized urban tree nursery shows no evident benefits to tree growth or mycorrhizal colonization » submitted to forests by Du et al. 2023.
Du et al. investigate the effects of a once-off compost tea application on growth and root mycorrhizal colonization of five urban tree species (Acer negundo, Corymbia maculata, Ficus platypoda, Hymenosporum flavum, Jacaranda mimosifolia) over six months. They found any benefit of compost tea application on mycorrhizal colonization and growth of trees in nursery except to A. negundo, where the treatment had significant negative effects on its growth.
Major comments:
This paper corresponds with scope of forests. However, I found many aspects of the manuscript which definitively need to be clarified particularly introduction, objectives, and materials and methods. The authors state that they are studying the effect of a single compost tea application to promote microbial communities in the growing substrate. Nevertheless, no results are presented regarding the impact of compost tea on microbial communities in the growing substrate. Therefore, the hypotheses formulated by the authors are not consistent with the main of the study. The major shortcoming of the paper is the impact of compost tea on substrate microbial communities is not considered. I suggest the authors rewrite the introduction and objectives. Another important point, which is not taken into account in the paper, would be to inoculate tree species with adapted AM and ECM fungal strains to urban soil conditions. Future research should include a detailed description of the nutrient needs for the 5 urban tree species in the presence of the main soil constraints and the role of native mycorrhizal fungal communities in nutrient acquisition.
Overall, I find the use of compost tea unconvincing based on the results obtained: no effect of compost tea on the growth and mycorrhizal colonization of tree species and sometimes negative effects that could be due to the introduction of unwanted compost tea microorganisms into the growing substrate. I did some other comments on materials and methods of the paper as follows:
Lines 35-36. Seven types of mycorrhizas are known according to Smith and Read (2008). Please correct.
Lines 44-52. The state of knowledge on the functioning of mycorrhizae in the 5 plant species and the main constraints in urban soils are lacking .
Lines 73-74. Why are chemical properties (i.e. pH, P and N available, OM) to the compost tea and growing substrate not determined?
Lines 73-74. Why was a single application of compost tea used? Response of tree species to a range of compost tea would be more appropriate.
Lines 89-90. I wonder if the number of replicates, five per treatment, is sufficient given the intraspecific variability that exists in tree species. Please clarify.
Lines 90-91. What are the proportions pine bark, coconut coir and sand in the growing substrate?
Lines 91-93. Why was A. negundo grown in pots of 27 L, while the other trees were grown in pots of 14 L? and why were different doses of tea compost applied?
Line 97. The sentence needs to be clarified
Lines 146-147. It is well known that ITS (ITS1, 5.8S and ITS2) and 18S are the appropriate markers to reveal the diversity of ECM and AM fungal communities, respectively. Why use ITS1?
Author Response
Dear reviewer,
Thank you for your time and patience reviewing our manuscript and providing valuable constructive comments. We have thoroughly discussed all the points and modified the manuscript accordingly. Please find our point-by-point response as follows. The line numbers listed here are based on the manuscript without track changes. Manuscript in pdf file with all tracked changes is also provided for your information.

Reviewer 2 Report
The authors present interesting topic of compost tea application in a containerized urban tree nursery and its potential to enhance plantlets growth and its mycorrhizal colonization. However, I have some questions and I would like authors to consider them.
Generally:
- scarcity of data: authors provide very scarce set of data - only stem height and diameter and the level of mycorrhizal colonization. For me it is questionable whether is there enough scientific merrit just to say - addition of compost tea stimulates or does not stimulate plant growth. Authors discuss compost tea chemical composition, but they do not show its composition (nutrients concentrations and/or their chemical forms or similar). Nor do they mention measurements of plant nutrition, plant elemental composition after amendment of the compost, in order to detect the compost influence on the plant nutrition. In some part authors say that tea compost is expected to have low macro and micronutrients levels. I wonder why do they then expect tea compost to have an impact on plant nutrition? Maybe due to some other biostimulating compounds besides the macro and micronutrients? But this is not explained.
- in Materials and methods authors do not mention how old the plantlets were at the begining of the experiment or their pheno phase.
- Was six months long enough period to detect expected results? In the Discussion I could not have found experiences of other researchers - how long is the expected time of the tea compost degradation and consequent release of nutrients or other stimulating compounds. Also, other researchers' experiences in how long period could we expect the detectable influence of tea compost on mycorrhizal induction.
- in Discussion authors comment the fact that usual use of N:P:K fertilizers does not stimulate mycorrhizal development or even supress it. I don't understand why did they not use reduced P content in the fertilizing process? Also, authors claim: "it is unlikely that any small macro- or micro-nutritional benefit from the compost tea application would have been detected beyond the larger inorganic fertilizer benefit provided to both control and treated groups. Similarly, it is likely that any positive nutrient contribution from any enhanced microbial mineralization processes would have also been masked by the larger inorganic fertilizer benefits." I wonder, why did authors then use this fertilization practice?
- authors also say that single application is hardly to have an effect. Why didn't they repeat it?
- authors made complex and expensive NGS analyses of microbial composition of the tea compost. I, personally, do not understand why they made it without any further explanation what does this composition say, what consequences does its composition have for the plantlets, how does it change during these six months ... What does this information on the compost microbial composition serve for? Why is it important for this experiment?
- lines 322-332: do these observations have anything to do with this experiment?
The questions I made are the main question that I have after reading this manuscript. Besides them, there are other minor suggestions, like:
- I would exclude Fig S2 - if I understand well, these are photos made by other authors and already available in the literature.
- I would inslude Table S1 into main text, because I think this is the main part of the experiment.
Etc. However, I'd like authors to give answers to the more important upper questions I made.
Author Response

(The authors gave the same response as above.)

Reviewer 3 Report
Authers in their study use Five common urban tree species to determine the effect of compost tea application on its growth and root mycorrhizal colonization over six months.
The concept of this research is interesting, and the paper generally is written clearly. However, there are some deficiencies and ambiguities in this text that should be corrected.
1- The references need updating; the manuscript contains only two references as of 2022.
2-Theauthourity of tree species should be written as follow”
Acer negundo L.
Corymbia maculata (Hook.) K.D.Hill & L.A.S.Johnson
Ficus platypoda (Miq.) A.Cunn. ex Miq.
Hymenosporum flavum F.Muell.
Jacaranda mimosifolia D.Don
2- A reference (s) is/are needed for Exotic/ Native tree species.
3- In the experimental design, authors should select trees of the same age, and at the same time, they should write their ages at the beginning of the experiment.
5- Physical and chemical characteristics for the used substrate should be provided.
6- - Six months experiment NOT enough for tree study.
7- Authors write that “Root samples were collected and processed four months after compost tea application”, Why four months?
Author Response

(The authors gave the same response as above.)

Round 2
Reviewer 2 Report
Authors replied to all my questions and comments in a clear way. I carefully read it. It is now clear that treated trees were grown according to the nursery's standard protocols which included the application of inorganic fertilizers in a routine way. Also, authors offered the justification for the experiment duration. However, I think that application of a compost tea in a containerized urban tree nursery and checking if it caused increased plant growth and their mycorrhizal colonization is not enough for a paper in a Q1 WoS journal, such as Forests. In my opinion, if this was accompanied with the additional analysis that could partly explain mechanisms that possibly took place in experiment, it would have stronger scientific aspect. I understand that authors followed normal practice of the advance tree nursery, but I think this is not the reason not to take additional aspects into consideration. I think this study is useful for the nurseries but potential paper is more suitable for some journal that is more oriented toward application and less toward research.
Author Response
Please find our response in the attached file.

Reviewer 3 Report
Dear authors Thank you for your responses to the comments.
Author Response
We thank reviewer 3 for the constructive comments.